# Assessment of standard HIV testing services delivery to injured persons seeking emergency care in Nairobi, Kenya: A prospective observational study

Adam R. Aluisio[1]*, Janet Sugut[2], John Kinuthia[3], Rose Bosire[4], Eric Ochola[3], Beatrice Ngila[3], Daniel K. Ojuka[5], J. Austin Lee[1], Alice Maingi[6], Kate M. Guthrie[7], Tao Liu[8], Mary Mugambi[9], David A. Katz[10], Carey Farquhar[10,11,12], Michael J. Mello[1]

1 Department of Emergency Medicine, Alpert Medical School of Brown University, Providence, RI, United States of America, 2 Department of Accident and Emergency, Kenyatta National Hospital, Nairobi, Kenya, 3 Department of Research & Programs, Kenyatta National Hospital, Nairobi, Kenya, 4 Center for Public Health Research, Kenya Medical Research Institute, Nairobi, Kenya, 5 Department of Surgery, University of Nairobi Faculty of Health Sciences, Nairobi, Kenya, 6 Department of Dermatology, Kenyatta National Hospital, Nairobi, Kenya, 7 Department of Psychiatry and Human Behavior, Alpert Medical School, Brown University, Providence, RI, United States of America, 8 Department of Biostatistics, Center for Statistical Sciences, Brown University School of Public Health, Providence, RI, United States of America, 9 Ministry of Health, Nairobi, Kenya, 10 Department of Global Health, University of Washington, Seattle, Washington, United States of America, 11 Department of Epidemiology, University of Washington, Seattle, Washington, United States of America, 12 Department of Medicine, University of Washington, Seattle, Washington, United States of America

* adam_aluisio@brown.edu

**Data Availability Statement:** Data are available at Zenodo DOI: 10.5281/zenodo.6498970.

## Abstract

Emergency departments (EDs) in Africa are contact points for key groups for HIV testing services (HTS) but understanding of ED-testing delivery is limited which may impeded program impacts. This study evaluated the offering and uptake of standard HTS among injured persons seeking ED care at Kenyatta National Hospital (KNH) in Nairobi, Kenya. An ED population of adult injured persons was prospectively enrolled (1 March—25 May 2021) and followed through ED disposition. Participants requiring admission were followed through hospital discharge and willing participants were contacted at 28-days for follow up. Data on population characteristics and HTS were collected by personnel distinct from clinicians responsible for standard HTS. Descriptive analyses were performed and prevalence values with 95% confidence intervals (CI) were calculated for HIV parameters. The study enrolled 646 participants. The median age was 29 years with the majority male (87.8%). Most ED patients were discharged (58.9%). A prior HIV diagnosis was reported by 2.3% of participants and 52.7% reported their last testing as >6 months prior. Standard ED-HTS were offered to 49 or 8.6% of participants (95% CI: 5.8–9.9%), among which 89.8% accepted. For ED-tested participants 11.4% were newly diagnosed with HIV (95% CI: 5.0–24.0%). Among 243 participants admitted, testing was offered to 6.2% (95% CI: 3.9–9.9%), with 93.8% accepting. For admitted participants tested 13.3% (95% CI: 4.0–35.1%) were newly diagnosed (all distinct from ED cases). At 28-day follow up an additional 22 participants reported completing testing since ED visitation, with three newly diagnosed. During the full

**Funding:** Author ARA and the overall work was supported by the National Institute of Allergy and Infectious Diseases at the National Institutes of Health (grant number: K23AI145411). The funders had no role in study design, data collection and analysis, decision to publish, or preparation of the manuscript.

**Competing interests:** The authors have declared that no competing interests exist.

follow-up period the HIV prevalence in the population tested was 10.3% (95% CI: 5.3–19.0%); all being previously undiagnosed. Offering of standard HTS was infrequent, however, when offered, uptake and identification of new HIV diagnoses were high. These data suggest that opportunities exist to improve ED-HTS which could enhance identification of undiagnosed HIV.

## Introduction

There are approximately 38 million people living with HIV (PLHIV) globally, of whom 95% reside in low-and middle-income countries (LMICs) and 70% in sub-Saharan Africa [1]. Although substantial progress has been made in diagnosing PLHIV, one in eight remain unaware of their status [2], and the populations of undiagnosed persons is increasing in some sub-Saharan Africa settings [3], The lack of testing and identification of PLHIV impedes epidemic control, a barrier which is potentiated by inadequate access to high-risk key populations who have not be well engaged by conventional approaches such as men, young people, people who inject drugs (PWID), men who have sex with men (MSM), and sex workers [4–7].

Emergency departments (ED) provide unscheduled yet frequent care to patients that may not otherwise regularly access health services [8–10]. In LMICs, episodic emergency care is particularly common for evaluation and treatment of injuries, which, similar to HIV, has greater likelihood of occurrence among key populations of men, young people, PWID, MSM and sex workers [11–15]. Additionally, data from sub-Saharan Africa has shown that patients seeking injury treatments have high burdens of HIV and are often first diagnosed during emergency care evaluations [16–19]. Although research from high-income countries demonstrates that ED patients are willing to engage in HIV programming [20–22], the delivery of provider-initiated testing and counseling (PITC) for HIV during emergency care in the sub-Saharan Africa context has been minimally studied [17, 23].

Kenya has a national HIV prevalence of 4.9% among people 15–64 years of age, with one in five unaware of their HIV disease [24]. Although Kenya national guidelines call for universal PITC at all healthcare interactions implementation has been met with challenges, especially among key groups [25–27]. As such, the Kenya Ministry of Health (MOH) has called for improved approaches for HIV testing services (HTS) in high-incidence populations [28]. Concurrently, Kenya experiences substantial injury burdens which also disproportionately occur in groups highly affected by HIV [15, 29, 30], indicating that ED-based HIV programming during commonly sought injury care could be an impactful interface for testing. Given the ED venue potential, this study aimed to evaluate the delivery and uptake of standard HIV testing by existing services and personnel among injured persons seeking emergency care in Kenya.

## Methods

### Ethics statement

Written informed consent was obtained from all enrolled participants. The informed consent process identified that the study was assessing HTS in the ED but did not mandate engagement for testing. The study was approved by the KNH ethics and research committee (P29/01/2020) and the institutional review board of Rhode Island Hospital (1501033–4).

## Study design, setting, and population

This prospective study was completed at Kenyatta National Hospital (KNH), the largest public health center in the nation of Kenya, located in the capital city of Nairobi. KNH maintains an ED that provides uninterrupted care access and has specialty services inclusive of surgical providers for injury treatments. The KNH ED has HTS available 24-hours a day with dedicated clinical space and staffing by personnel trained in Voluntary Counseling and Testing (VCT) for HIV at all times tasked with delivering PITC to ED patients. Kenya national guidelines stipulate that universal opt-out PITC for HIV should be offered at every healthcare encounter, and as such, HTS in the KNH ED are provided free of charge using Alere Determine HIV-1/2 assays. ED-based HTS are provided by the VCT providers who can screen patients independently or be provided real-time testing referrals from ED clinical personnel at their discretion. Standardized records for all patients undergoing HIV testing are maintained using national reporting documentation and procedures, and any person diagnosed with HIV is provided follow up treatment services at MOH facility if willing [25].

All adult patients (≥18 years of age), presenting for injury care to the ED study site were eligible for participation. Enrollment was carried out in the ED between 1 March and 25 May 2021. Patients known to be pregnant, prisoners of the state and those unable or unwilling to provide informed consent were excluded. Patients with altered mental status regardless of the etiology were excluded. Injury designation was based on the standardized triage classification used in the study setting [30], by clinical staff who were independent of the research personnel. Guidelines for Strengthening the Reporting of Observational Studies in Epidemiology were followed [31].

## Data collection and management

After the provision of informed consent, trained study personnel, who were present in the ED 24-hours a day during the enrollment phase and independent of the clinical staff, collected standardized data using digital devices and online databases [32]. Consenting participants had enrollment data collected as close to ED arrival as possible without impeding clinical care. Baseline data included information on sociodemographic aspects, past medical history, characteristics of the injury event, previous health behaviors pertaining to HIV testing and prevention, and use of alcohol and other recreational substances. Participants were followed until ED disposition (completion of ED care) at which time they had information collection on standard HIV testing engagement (offering by clinical staff and uptake by the participant). Data on injury burdens and psychometric data on participant perspectives on ED-based HIV services were also collected at time of ED disposition. Injury characteristics were classified via the standardized Abbreviated Injury Scale, as has been used previously in similar settings [33, 34]. The psychometric data utilized a battery of Likert items which were piloted for understandability and reproducibility in the study setting, and assessed with five-point response scales. The items focused on appropriateness of ED-HIV testing and specific aspects of physical space, interaction times, ability to ensure confidentiality and adequacy of patient-provider relationships (S1 Table).

Participants with an ED disposition of admission as inpatients were followed until time of discharge from the hospital or through 28-days post-enrollment, if they remained hospitalized at that time. The discharge assessment evaluated inpatient HIV testing engagement (offering by clinical staff and uptake by the participant). At time of enrollment all participants were asked if they would be willing to be contacted at 28-days from the index ED visit for phone follow up. If they assented the participants were contacted at that timepoint (within a five-day window) and had data collected on interval HIV services engagement. For participants that

remained admitted as inpatients at the 28-day timepoint this assessment was omitted. Participants who died during facility-based care had data adjudicated from medical documentation when possible.

### Assessment of HIV testing services

Standard ED-based HIV test offering was assessed at time of disposition based on patient report. For participants who reported accepting HIV testing, the receipt and outcome of testing was cross-validated using the MOH national reporting documentation for HTS at the study site. For those that reported not completing ED testing, the medical documentation was reviewed to evaluate for any unreported testing encounters. At inpatient discharge, data collection on HIV test offering and uptake via patient report was cross-checked via the medical documentation and testing registries. At 28-day follow up participants were evaluated for any interval HIV testing, outcomes and follow up care since their hospital treatments. As the study evaluated standard HTS delivery and uptake by existing services and personnel in the study setting, testing services were not offered by study staff who were independent of the clinical providers

### Statistical analysis

Data analysis was completed using STATA version 16.0 (College Station, USA). The primary outcome of interest was offering of standard PITC for HIV in the ED setting among the population of injured persons by the existing clinical providers and services. A sample size of 624 participants was calculated to achieve the primary outcome of identifying the prevalence of ED-based HIV test offering with a precision of 3.4% (based on one-half the width of the binomial confidence interval) [35]. Secondary outcomes included completion of ED-based PITC, offering and completion of inpatient HTS and the identification of newly diagnosed PLHIV in the study population.

Descriptive analyses were undertaken for the enrolled population using frequencies with percentages or medians with associated interquartile ranges (IQR) as appropriate. Likert data were analyzed using medians with IQRs and frequencies represented by response density for individual items across the outcome scale. Outputs of prevalence with associated 95% confidence intervals (CI) were calculated for the HIV parameters of interest and stratified for the ED and inpatient care periods, for the 28-day timepoint and for overall follow up (composite of ED, inpatient and 28-day assessment). Two participants tested in the ED, and found to have HIV, underwent repeat testing during inpatient care. For those participants testing outcomes for only the ED testing events were used in analyses. No participants completing HIV testing at the 28-day follow up time point were tested during facilities-based care (defined as the composite of ED and inpatient).

Participant characteristics base on either ED or facilities-based offering of HIV testing were compared using Pearson $X^2$ or Fisher's exact tests for categorical variables and by Mann-Whitney tests for continuous variables. In the comparative analyses, a statistical correction was used to set an appropriate significance cut-off (p<0.004) [36]. An *a priori* secondary analysis using multivariable logistic regression to identify factors associated with ED-based as well as facilities-based offering of HTS was planned, however, the data were insufficient to support those analyses.

## Results

During the enrollment period, 1,282 patients presenting for injury care to the ED setting were screened for participation. Among those, 73 (5.7%) declined participation and 563 (43.9%) did

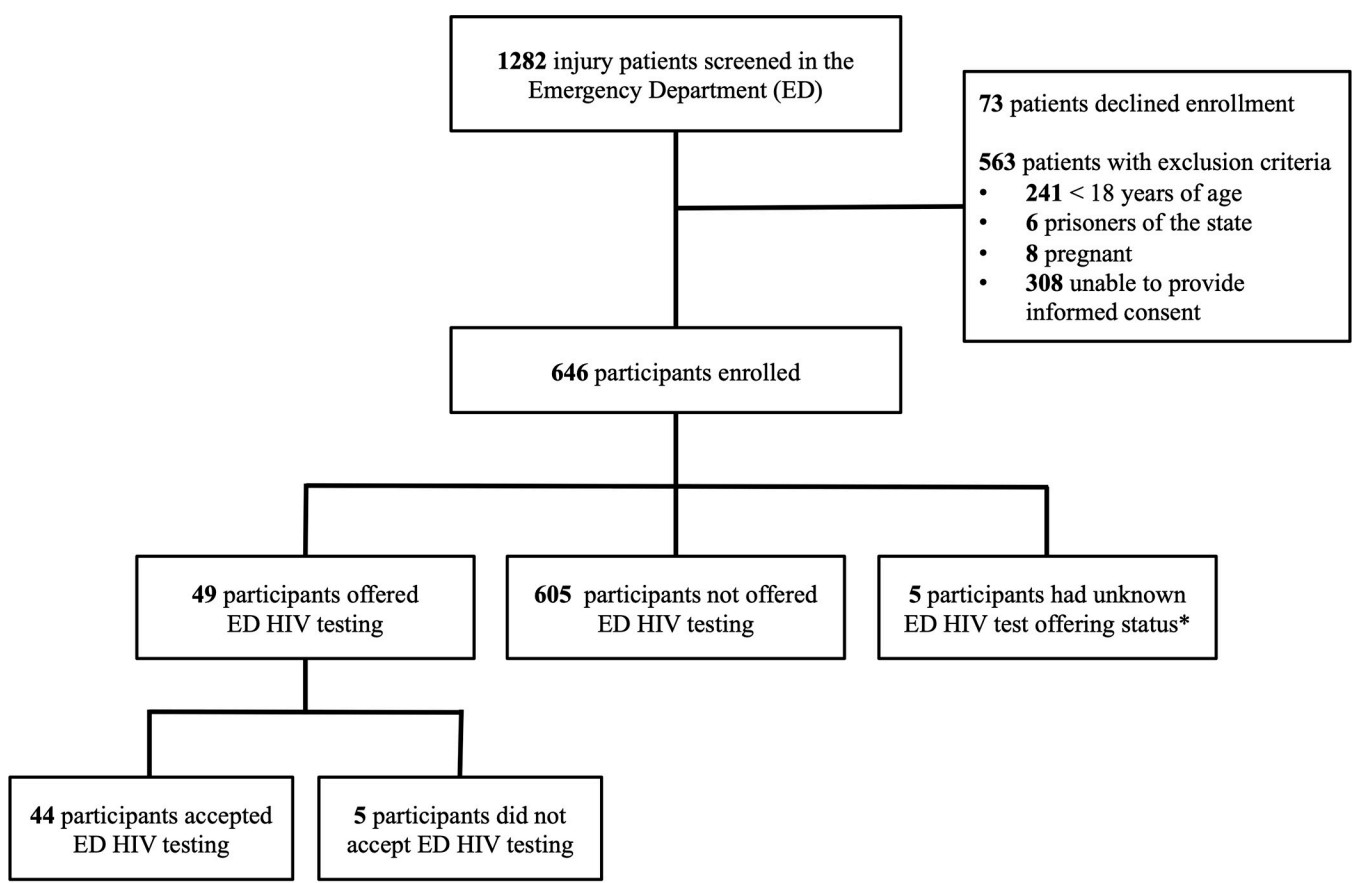

*1 participant died during ED care, 2 participants lost to follow-up from the ED and 2 participants with missing data on HIV test offering evaluation

**Fig 1. Study population.**

not meet inclusion criteria. A total 646 participants were enrolled, of whom 641 had complete data for the primary outcome (Fig 1). Within the enrolled study population 270 participants (41.8%) were admitted for inpatient care from the ED, of which 243 had complete inpatient follow up data (four participants died and 23 were lost to follow-up). Contact assent for 28-day follow up was provided by 553 participants, and of those, 443 completed data collection.

## Characteristics of the study population

Within the population there was a male predominance (87.8%). Participant median age was 29 years (IQR: 25, 37 years), with 30.9% being $\leq$ 25 years old. The majority of those enrolled had no established primary care provider (53.9%) and 6.7% of the participants reported having at least one chronic medical condition. Alcohol use was reported by 48.6% of the cohort and condom use during sexual intercourse was reported as never by 51.1% of respondents. Prior HIV testing was reported by 86.7% participants, with the majority (52.7%) reporting their most recent testing occurring more than six months prior to enrollment. For those previously tested, 2.3% reported being PLHIV (Table 1). The reported location for the last HIV testing events was MOH facilities in 62.3% and private clinics in 27.3%, with 13 participants (2.0%) reporting using HIV self-tests.

**Table 1. Participant characteristics.**

| Variable | n (%) or Median (IQR) |
|---|---|
| *Sex* | |
| Male | 567 (87.8%) |
| Female | 79 (12.2%) |
| *Age (years)* | 29 (25, 37) |
| *Employment Status* | |
| Not working | 116 (18.0%) |
| Laborer | 323 (50.1%) |
| Self-employed | 129 (19.9%) |
| Professional | 78 (12.0%) |
| *Relationship Status* | |
| Single | 217 (33.6%) |
| Married | 301 (46.6%) |
| In a relationship | 77 (11.9%) |
| Divorced / Widowed | 43 (6.7%) |
| Wishes not to disclose | 4 (0.6%) |
| Missing / Unknown | 4 (0.6%) |
| *Education* | |
| Completed primary schooling (or less) | 252 (39.0%) |
| Completed secondary schooling | 263 (40.7%) |
| Greater than Secondary schooling | 131 (20.3%) |
| *Reported Chronic Medical Condition* | |
| Yes | 43 (6.7%) |
| No | 603 (93.3%) |
| *Has Primary Care Provider* | |
| Yes | 297 (46.0%) |
| No | 348 (53.8%) |
| Missing / Unknown | 1 (0.2%) |
| *Uses Recreational Alcohol* | |
| Yes | 314 (48.6%) |
| No | 323 (50.0%) |
| Missing / Unknown | 9 (1.4%) |
| *Uses Recreational Substances (non-alcohol)[†]* | |
| Yes | 141 (21.8%) |
| No | 505 (78.2%) |
| *Frequency of Condom use during intercourse* | |
| Always | 114 (17.7%) |
| Sometimes | 142 (21.9%) |
| Never | 330 (51.1%) |
| Wishes not to disclose | 14 (2.2%) |
| Missing | 46 (7.1%) |
| *Previously Tested for HIV* | |
| Yes | 560 (86.7%) |
| No | 83 (12.8%) |
| Missing | 3 (0.5%) |
| *Timing of Last HIV Testing[‡]* | |
| ≤ 6 months prior | 265 (47.3%) |
| > 6 months prior | 295 (52.7%) |

(*Continued*)

**Table 1.** (Continued)

| Variable | n (%) or Median (IQR) |
|---|---|
| Prior HIV Test Results# | |
| Positive | 13 (2.3%) |
| Negative | 529 (94.5%) |
| Wishes not to disclose | 6 (1.1%) |
| Missing | 12 (2.1%) |

†, Included substances were classified as depressant, psychogenic or stimulant

‡, Denominator = 560, number of participants reporting prior HIV testing

The most common mechanisms of injury were road traffic accidents (51.9%) and falls (15.3%). The most frequent anatomical region of injury was craniofacial (51.1%) and lower extremities 47.6%. The median duration of ED care was 4.2 hours (IQR: 2.5, 6.0 hours). The majority of participants (55.9%) were discharged from the ED and admission for inpatient care occurred in 41.8% of cases (Table 2).

## HIV testing services

Standard ED-based HTS from existing clinical providers were offered to 49 or 8.6% of enrolled participants (95% CI: 5.8–9.9%). When offered, HTS in the ED were accepted by 89.8% of participants (95% CI: 78.2–95.6%). A new diagnosis of HIV was found in 11.4% of ED-tested participants (95% CI: 5.0–24.0%). No participants who were previously known PLHIV completed ED-based testing. For the 243 participants admitted from the ED with completed follow up, 14 or 6.2% were offered testing during inpatient care (95% CI: 3.9–9.9%). Among those offered inpatient testing, 93.8% accepted (95% CI: 62.4–99.3%) and 13.3% of those tested were found to have HIV (95% CI: 2.9–44.0%). Among the 443 participants with completed 28-day follow up, 22 (5.0%, 95% CI: 3.3–7.4%) reported completing HIV testing since their index ED encounter. Three of these participants (13.0%, 95% CI: 4.0–35.1%) were newly diagnosed with HIV, and all had initiated treatment. During the full follow-up period the identified prevalence of HIV in the population tested was 10.3% (95% CI: 5.3–19.0%), all being previously undiagnosed PLHIV.

In comparative analyses between those offered and not offered ED-HTS no significant differences in characteristics were identified (Table 3). In evaluation of factors associated with overall facilities-based HIV test offering, only relationship status was significantly different among variables assessed, with those not offered testing more commonly reporting their status as single at 34.3% as compared to those offered at 27.5% (p<0.001) (S2 Table).

## Perspectives on ED-based HIV care

In Likert item data participants reported high agreement with the need for universal offering of ED-based HTS (median = 5, IQR: 4,5) and for test offering specifically for patients seeking emergent injury care (median = 5, IQR: 3,5). Likert item data on the suitability of HIV testing in relation to aspects of ED physical space, interaction times, ability to ensure confidentiality and adequacy of patient-provider relationships demonstrated that the majority of participants responded with the highest level of agreement across all items (≥60.0%). There were however varying distributions of agreement among a minority of participants for suitability of physical space with 11.2% indicating the lowest level of agreement. Also, for adequacy of confidentiality

**Table 2. Clinical characteristics and outcomes.**

| Variable | n (%) or Median (IQR) |
|---|---|
| *Mechanism of Injury* | |
| Road Traffic Injury | 335 (51.9%) |
| Electrocution | 10 (1.6%) |
| Fall | 99 (15.3%) |
| Gun Shot Wound | 2 (0.3%) |
| Burn | 13 (2.0%) |
| Stab or Cut | 93 (14.4%) |
| Intimate partner violence | 5 (0.8%) |
| Other blunt | 43 (6.7%) |
| Other Penetrating | 16 (2.4%) |
| Missing / Unknown | 30 (4.6%) |
| *Transferred from Another Health Facility* | |
| Yes | 272 (42.1%) |
| No | 372 (57.6%) |
| Missing | 2 (0.3%) |
| *Time from Injury to Presentation (hours)* | 12.5 (5, 24.5) |
| *Body Region(s) Injured*[*] | |
| Craniofacial | 328 (51.1%) |
| Neck | 42 (6.6%) |
| Thorax | 57 (8.9%) |
| Abdomen-pelvis | 40 (6.4%) |
| Cervical Spine | 8 (1.2%) |
| Thoracic Spine | 2 (0.3%) |
| Lumbar Spine | 8 (1.3%) |
| Upper extremity | 237 (37.4%) |
| Lower extremity | 304 (47.6%) |
| Genitalia | 5 (0.8%) |
| *Types of Injuries Sustained*[†] | |
| Open wound | 404 (62.5%) |
| Bruise | 607 (93.9%) |
| Dislocation | 60 (93%) |
| Fracture Open | 108 (16.7%) |
| Fracture Closed | 223 (34.5%) |
| Burn | 51(7.9%) |
| Pneumothorax | 4(0.6%) |
| Hemothorax | 2(0.3%) |
| Vascular Injury | 21(3.3%) |
| Nerve Injury | 9(1.4%) |
| Hemoperitoneum | 3(0.4%) |
| Organ Injury | 23(3.5%) |
| *Duration of Emergency Care (hours)* | 4.2 (2.5, 6.0) |
| *Emergency Department Disposition* | |
| Discharge | 361(55.9%) |
| Admitted to Inpatient Service | 258 (39.9%) |
| Admitted to Operating Theater | 12 (1.9%) |
| Transferred to another facility | 7 (1.1%) |
| Left Prior to Care Completion | 7 (1.1%) |

(*Continued*)

**Table 2.** (Continued)

| Variable | n (%) or Median (IQR) |
|---|---|
| Deceased | 1 (0.1%) |

†, Due to potential for multiple observations per participant sum may exceed 100%

and patient-provider relationships lower levels of agreement were reported by 12.4% and 10.1% of participants, respectively (Fig 2).

## Discussion

This study contributes to the developing literature on HTS in ED settings, and represents some of the first prospective data with longitudinal assessment of persons presenting for care of injuries in sub-Saharan Africa [17, 23]. The results demonstrate that there exists substantial potential to increase HTS for those seeking emergency care of injuries, and that the population had frequent uptake of testing when it was offered in the study setting. Given these findings, in conjunction with the relatively high frequency of newly identified PLHIV among those completing testing, development of improved ED-based HIV-programming may represent an impactful approach to enhance testing and identification of undiagnosed individuals from high-incidence populations.

In the current data standard ED-based PITC for HIV by existing clinical providers was infrequently offered to those seeking care. This is congruent with the limited data that exists from other emergency care settings in sub-Saharan Africa [17, 18]. In a 2018 cohort from South Africa, in which the majority of patients were men and seeking emergency injury treatments, only 25% of patients were offered testing [18]. Given that Kenya national guidelines call for universal opt-out PITC at every healthcare encounter [25], and that resources to support this exist in the clinical setting evaluated the low occurrence of test offering in the studied population indicates that there is potential to increase testing delivery from the ED setting. Furthermore, as emergency care interactions in LMICs are common, with a median of 30,000 visits per facility per year [9], opportunities may exist to augment HTS to large numbers of persons from LMIC ED venues. As injuries are one of the most frequent reasons for receiving emergency care, and those most likely to be injured are also key target populations for testing initiatives the ED venue may be an even more important focus point to improve HIV service delivery [6, 11–14]. In Kenya specifically, younger people, who are more frequently injured and also comprise more than half of all new HIV infections are also less likely to test, which inhibits success of HIV control programs and further points to the potential impacts of accessing this group via ED-based HIV care [27, 28]. In light of the unique systems, environmental and human aspects inherent in emergency care, coupled to the observed infrequent offering of HIV testing by the existing services in the current data, and prior studies [17, 18], research to better understand approaches for improved ED-HTS in Kenya and broader sub-Saharan Africa settings are needed.

When offered, acceptance of HIV testing in the population was high, both in the ED and inpatient care venues. Furthermore, the majority of participants strongly agreed that HTS should be provided during emergency care and that aspects of the environment and care interactions are suitable for ED-based testing. These data affirm similar findings from previous studies reporting frequent acceptance of HIV testing in emergency and acute care populations in sub-Saharan Africa when testing is offered [18, 19, 26]. This suggests that the challenges to ED-based HTS delivery may be driven with a greater relative predominance by systems and

**Table 3.** Characteristics stratified by emergency department offering of HIV testing services.

| Variable | HIV Testing Services Offered (n = 49) n (%) or Median (IQR) | HIV Testing Services Not Offered (n = 592) n (%) or Median (IQR) | P value |
|---|---|---|---|
| *Sex* | | | |
| Male | 38 (77.6%) | 525(88.7%) | 0.064 |
| Female | 11 (22.5%) | 67(11.3%) | |
| *Age (years)* | | | |
| 18-25yrs | 15 (30.6%) | 178 (30.1%) | 0.578 |
| 26-44yrs | 31 (63.3%) | 346 (58.5%) | |
| > 44yrs | 3 (6.1%) | 68 (11.5%) | |
| *Relationship Status* | | | |
| Single | 18 (36.7%) | 197 (33.5%) | 0.014 |
| Married | 19 (38.8%) | 279 (47.4%) | |
| In a relationship | 6 (12.2%) | 71 (12.1%) | |
| Divorced / Widowed | 3 (6.1%) | 40 (6.8%) | |
| Wishes not to disclose | 3 (6.3%) | 1(0.2%) | |
| Missing / Unknown | 0 (0.0%) | 1(0.2%) | |
| *Education* | | | |
| Completed primary schooling (or less) | 26 (53.1%) | 223 (37.7%) | 0.003 |
| Completed secondary schooling | 21 (42.9%) | 241 (40.7%) | |
| Greater than Secondary schooling | 2 (4.1%) | 128 (21.6%) | |
| *Has Primary Care Provider* | | | |
| Yes | 26 (53.1%) | 270 (45.6%) | 0.032 |
| No | 22 (44.9%) | 322 (54.4%) | |
| Missing / Unknown | 1 (2.0%) | 0 (0.0%) | |
| *Uses Recreational Alcohol* | | | |
| Yes | 23 (47.9%) | 288 (49.2%) | 0.891 |
| No | 25 (52.1%) | 296 (50.6%) | |
| Missing / Unknown | 1 (2.0%) | 1 (0.2%) | |
| *Uses Recreational Substances (non-alcohol)[†]* | | | |
| Yes | 7 (14.3%) | 133 (22.5%) | 0.183 |
| No | 42 (85.7%) | 459 (77.5%) | |
| *Timing of Last HIV Testing* | | | |
| ≤ 6 months prior | 22 (44.9%) | 241 (40.7%) | 0.567 |
| > 6 months prior | 27 (55.1%) | 351 (59.3%) | |
| *Frequency of Condom use during intercourse* | | | |
| Always | 6 (12.2%) | 107 (18.07%) | 0.479 |
| Sometimes | 12 (24.5%) | 128 (21.6%) | |
| Never | 24 (49.0%) | 304 (51.4%) | |
| Wishes not to disclose | 2 (4.1%) | 12 (2.0%) | |
| Missing | 5 (10.2%) | 41 (6.9%) | |
| *Number of Body regions injured* | 1 (1, 2) | 1 (1, 2) | 0.424 |
| *Transferred from Another Health Facility* | | | |
| Yes | 13 (26.5%) | 258 (43.6%) | 0.050 |
| No | 36 (73.5%) | 332 (56.1%) | |
| Missing | 0 (0.0%) | 2(0.3%) | |
| *Emergency Department Arrival Time* | | | |

(*Continued*)

**Table 3.** (Continued)

| | HIV Testing Services Offered (n = 49) | HIV Testing Services Not Offered (n = 592) | |
|---|---|---|---|
| Variable | n (%) or Median (IQR) | n (%) or Median (IQR) | P value |
| Day (6am– 6pm) | 34 (69.4%) | 323 (54.5%) | 0.045 |
| Night (6pm– 6 am) | 15 (30.6%) | 269 (45.4%) | |
| *Emergency Department Disposition* | | | |
| Discharge | 28 (57.14%) | 331 (56.0%) | 0.756 |
| Admitted to Inpatient Service | 19 (38.8%) | 239 (40.4%) | |
| Admitted to Operating Theater | 1 (2.0%) | 11 (1.9%) | |
| Transferred to another facility | 1 (2.0%) | 5 (0.9%) | |
| Left Prior to Care Completion | 0 (0.0%) | 1 (0.2%) | |
| Deceased | 0 (0.0%) | 0 (0.0%) | |

†, Included substances were classified as depressant, psychogenic or stimulant

provider factors such as insufficient resources, time limitations or and perceived testing needs rather than those of the patient recipients. However, due to the complex elements surrounding HIV testing during emergency care, future studies to yield more comprehensive data on opportunities and challenges in ED-based HIV care are needed to inform development of appropriate and effective implementation approaches for service enhancement.

The proportion of newly identified PLHIV among participants completing testing was more than two-fold greater than the baseline population prevalence in the study setting [24]. These data comport with prior studies in which HIV burdens among persons with injuries in Africa have been found to be greater than the prevalence in the corresponding populations

| Likert Item Topical Focus* | n | Median (IQR) | Response Frequency n (%) | | | | | Density Scale |
|---|---|---|---|---|---|---|---|---|
| | | | *1 (Do not agree at all)* | *2 (Agree a little)* | *3 (Agree somewhat)* | *4 (Agree a lot)* | *5 (Agree completely)* | |
| Testing should be offered for all patients | 629 | 5 (4, 5) | 50 (8.0%) | 48 (7.6%) | 46 (7.3%) | 64 (10.2%) | 421 (66.9%) | Lesser Proportion |
| Testing should be offered for injured patients specifically | 639 | 5 (3, 5) | 42 (6.6%) | 45 (7.0%) | 80 (12.5%) | 51 (8.0%) | 421 (65.9%) | |
| The physical space and room are acceptable for testing | 571 | 5 (3, 5) | 64 (11.2%) | 52 (9.1%) | 67 (11.7%) | 36 (6.3%) | 352 (61.7%) | |
| Confidentiality and trust are sufficient for testing | 575 | 5 (3, 5) | 42 (7.3%) | 71 (12.4%) | 79 (13.7%) | 38 (6.6%) | 345 (60.0%) | |
| Patient-provider relationships are adequate for testing | 591 | 5 (3, 5) | 16 (2.7%) | 60 (10.1%) | 99 (16.8%) | 53 (9.0%) | 363 (61.4%) | |
| The treatment time a patient has is enough for testing | 626 | 5 (4, 5) | 8 (1.3%) | 21 (3.4%) | 83 (13.2%) | 52 (8.3%) | 462 (73.8%) | Greater Proportion |

* Full text Likert items provided in Supplement 1

**Fig 2. Patient perspectives on emergency department HIV testing.**

[16], and that ED-based testing frequently identifies incident infections [18, 19, 37]. As the current data pertaining to testing outcomes were limited, secondary to the low numbers of test offered, precise estimates on HIV burdens in the population studied are difficult to make and larger sampling frames are required. However, the high observed disease burden does support the potential to increase identification of PLHIV through engagement and testing in ED settings. Although gains in combating the HIV epidemic require care coordination and treatment following diagnosis [38], identification of PLHIV is a requisite first step, which could better reached with strengthened ED-HTS in sub-Saharan Africa, an approach which has been recognized as important in alternative higher-resourced settings [39].

The majority of injury care globally is completed during emergency/acute care encounters and does not require inpatient management [15, 40]. This was observed in the current study population, however a substantial minority of participants did require admission. Among the admitted participants, the data on HIV testing was similar to the ED venue, with testing being infrequently offered but when offered commonly accepted. This may indicate that challenges to HIV testing among injured persons is not isolated only to the emergent care period specifically in the ED setting, and better understanding of how to address these challenges throughout the continuum of treatment to enhance HTS would be instructive in future program development.

## Limitations

This study must be interpreted with consideration of certain limitations. Even though prospective methods were used there was a small amount of missing data for the primary outcome on ED-based HIV testing. As the amount of missing data was minimal, this is unlikely to have significantly impacted the findings. However, there was a larger degree of missing data for inpatient and 28-day follow up assessments, such that outcomes for those time-points may be less certain. The inclusion criteria requiring participant consent without the option for consent by an alternative proxy may have introduced selection bias as those with persistent clinical instability or altered mental status were excluded. Although this could affect the generalizability, the internal validity pertaining to the population of interest would not likely be impacted. Additionally, as the participants were aware, via the informed consent process, that the research was evaluating HTS they could have been more willing to accept testing when offered, which could have resulted in overestimation of the HTS acceptance outcomes. The data may not be completely generalizable to alternative settings as they are derived from a clinical site that is relatively well-resourced for the provision of HIV testing and further study is needed to characterize HIV services in settings with differential resources. Related, although an *a priori* secondary objective was to assess factors associated with the standard offering of HIV testing in the population the low event rate for test offering precluded the ability to complete those analyses. As such, future research aiming to assess opportunities and challenges associated with ED-based HIV testing in high-burden settings, such as Kenya, will be beneficial to inform programming. Finally, the study was completed during the pandemic caused by SARS CoV-2 and it is reasonable that transmission concerns could have reduced the delivery of HTS in the study setting as has been observed in other research [41]. The study is however valid within the context in which it was completed, and as pandemic conditions are likely to persist, particularly in LMICs where HIV burdens have the greatest impacts [1, 42], the results provide relevant data to inform HTS in the contemporary global context.

## Conclusions

This prospective longitudinal data found that the offering and delivery of ED-based HTS to the population of adult injured persons seeking care was low, but when offered, there was

frequent uptake and a high relative identification of newly diagnosed PLHIV as compared the prevalence in the general population. These findings suggest that there are opportunities to increase ED-HTS delivery, and that if achieved, this could enhance identification of undiagnosed persons during emergency care encounters. Given the potential margin for gains in HIV testing and diagnosis, coupled to the limited available research from LMIC emergency care settings, future studies to better understand of opportunities and challenges in ED-based HIV programming are needed to inform development of more effective and context appropriate testing strategies.

## Supporting information

**S1 Table. Likert items and topical focus.**
(DOCX)

**S2 Table. Characteristics stratified by facilities-based offering of HIV test services.**
(DOCX)

**S1 Text. Inclusivity in global research.**
(DOCX)

## Acknowledgments

The authors thank all of the personnel who assisted in the research and study participants who took part in the work. As well, acknowledgment to Ananya Suram who contributed to production of data graphics.

## Author Contributions

**Conceptualization:** Adam R. Aluisio, John Kinuthia, Rose Bosire, Daniel K. Ojuka, Alice Maingi, Kate M. Guthrie, Mary Mugambi, Carey Farquhar.

**Data curation:** Adam R. Aluisio, Janet Sugut, John Kinuthia, Rose Bosire, Beatrice Ngila, J. Austin Lee, Tao Liu, Michael J. Mello.

**Formal analysis:** Adam R. Aluisio, John Kinuthia, Eric Ochola, Beatrice Ngila, J. Austin Lee, Tao Liu, Mary Mugambi, David A. Katz, Carey Farquhar, Michael J. Mello.

**Funding acquisition:** Adam R. Aluisio, John Kinuthia, Rose Bosire, Alice Maingi, Kate M. Guthrie, Tao Liu, Mary Mugambi, David A. Katz, Carey Farquhar, Michael J. Mello.

**Investigation:** Adam R. Aluisio, Janet Sugut, John Kinuthia, Rose Bosire, Eric Ochola, Daniel K. Ojuka, J. Austin Lee, Alice Maingi, Kate M. Guthrie, Tao Liu, Mary Mugambi, David A. Katz, Carey Farquhar, Michael J. Mello.

**Methodology:** Adam R. Aluisio, Janet Sugut, John Kinuthia, Eric Ochola, Beatrice Ngila, Daniel K. Ojuka, J. Austin Lee, Alice Maingi, Kate M. Guthrie, Tao Liu, Mary Mugambi, David A. Katz, Carey Farquhar, Michael J. Mello.

**Project administration:** Adam R. Aluisio, Janet Sugut, John Kinuthia, Rose Bosire, Eric Ochola, Beatrice Ngila, Daniel K. Ojuka, J. Austin Lee, Alice Maingi, Kate M. Guthrie, Mary Mugambi, David A. Katz, Carey Farquhar, Michael J. Mello.

**Resources:** Adam R. Aluisio, Janet Sugut, John Kinuthia, Eric Ochola, Beatrice Ngila, Daniel K. Ojuka, Kate M. Guthrie, Tao Liu, Mary Mugambi, David A. Katz, Carey Farquhar, Michael J. Mello.

**Software:** Eric Ochola.

**Supervision:** Adam R. Aluisio, Janet Sugut, John Kinuthia, Rose Bosire, Beatrice Ngila, J. Austin Lee, Alice Maingi, Kate M. Guthrie, Tao Liu, David A. Katz, Carey Farquhar, Michael J. Mello.

**Validation:** Adam R. Aluisio, Janet Sugut, John Kinuthia, Rose Bosire, Eric Ochola, Beatrice Ngila, Tao Liu, Carey Farquhar.

**Visualization:** Adam R. Aluisio, Eric Ochola.

**Writing – original draft:** Adam R. Aluisio, Janet Sugut, John Kinuthia, Rose Bosire, Eric Ochola, Beatrice Ngila, Daniel K. Ojuka, J. Austin Lee, Alice Maingi, Kate M. Guthrie, Tao Liu, Mary Mugambi, David A. Katz, Carey Farquhar, Michael J. Mello.

**Writing – review & editing:** Adam R. Aluisio, Janet Sugut, John Kinuthia, Rose Bosire, Eric Ochola, Beatrice Ngila, Daniel K. Ojuka, J. Austin Lee, Alice Maingi, Kate M. Guthrie, Tao Liu, Mary Mugambi, David A. Katz, Carey Farquhar, Michael J. Mello.

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
