## [Decision Letter · Decision Letter 0]

2 Sep 2022

PGPH-D-22-00728

Assessment of Standard HIV Testing Services Delivery to Injured Persons Seeking Emergency Care in Nairobi, Kenya: A Prospective Observational Study

Dear Dr. Aluisio,

Thank you for submitting your manuscript to PLOS Global Public Health. After careful consideration, we feel that it has merit but does not fully meet PLOS Global Public Health’s publication criteria as it currently stands. Therefore, we invite you to submit a revised version of the manuscript that addresses the points raised during the review process.

We look forward to receiving your revised manuscript.

Kind regards,

Sabine Hermans

Academic Editor

Journal Requirements:

2. Please amend your online detailed Financial Disclosure statement. This is published with the article. It must therefore be completed in full sentences and contain the exact wording you wish to be published.

3. Please update your online Competing Interests statement. If you have no competing interests to declare, please state: “The authors have declared that no competing interests exist.”

4. We have noticed that you have uploaded Supporting Information files, but you have not included a list of legends. Please add a full list of legends for your Supporting Information files after the references list.

5. We do not publish any copyright or trademark symbols that usually accompany proprietary names, eg (R), (C), or TM  (e.g. next to drug or reagent names). Please remove all instances of trademark/copyright symbols throughout the text, including ™ on page 5.

Additional Editor Comments (if provided):

This is an important prospective study showing the opportunity of HIV testing in emergency care settings in sub-Saharan Africa. Please find some minor comments for your attention by the reviewers below.

In addition, I had two comments/questions:

- Could you maybe include whether this project was part of a larger prospective study? I.e., did the participants give informed consent to a study about HTS in the emergency department, or to a larger study on additional outcomes? I can imagine if it were the former, those consenting might be more inclined to accept HTS when offered (but conversely it would seem odd that HTS was offered so infrequently by the staff if they knew a study evaluating this in their unit was ongoing). Therefore it makes more sense that these outcomes were embedded in a larger project looking at emergency care in general. This would be useful information to include somewhere.

- Could you maybe speculate why the offering of HTS was so low?

Reviewers' comments:

Reviewer's Responses to Questions

**Comments to the Author**

1. Does this manuscript meet PLOS Global Public Health’s publication criteria? Is the manuscript technically sound, and do the data support the conclusions? The manuscript must describe methodologically and ethically rigorous research with conclusions that are appropriately drawn based on the data presented.

Reviewer #1: Yes

Reviewer #2: Yes

2. Has the statistical analysis been performed appropriately and rigorously?

Reviewer #1: Yes

Reviewer #2: Yes

3. Have the authors made all data underlying the findings in their manuscript fully available (please refer to the Data Availability Statement at the start of the manuscript PDF file)?

Reviewer #1: Yes

Reviewer #2: Yes

4. Is the manuscript presented in an intelligible fashion and written in standard English?

Reviewer #1: Yes

Reviewer #2: Yes

5. Review Comments to the Author

Reviewer #1: The paper is important for better understanding of an important public health potential to increase access to HIV testing. To understand better the contextual situation at this particular one study site it would be important to know how alcohol use prior to presentation at ED may or may not have influenced consent and testing procedures. More than half of the study population admitted to social drinking, majority of reasons for attending ED were accidents and falls which is kmown to have alcoho; as a contributing factor. Exclusion to study population were not detailed enough to understand better if consiousness was seriously affected by trauma or alcohol or emergency conditions to fullfill eligibility criteria.

Large majority of study population knew already their serostatus, almost half from a recent test less than 6 months ago which is high as compared to other urban areas in the region which may affect consent and could be elaborated on in the discussion

It was not that clear from the data presentation if during the 2-6 hours of stay at the ED, most or all had given consent or that consent was provided later during admission for those half requiring admission.

The conclusion to expand PITC during ED should take into account a point earlier made in the paper that the HIV service provision at KNH is exceptional and 24 hrs present and a call for making available those services with trained staff and 24hrs easily accessible supplies to be made stronger.

Reviewer #2: Dear authors,

Thank you for completing this rigorous and important research. Your study design, results, data analysis and conclusions appear technically sound. You have discussed your limitations well (ie inability to compare the characteristics of the cohort who declined testing (N only 4)).

I have a few minor questions and optional suggestions.

What were the main reasons that 308 of the 563 patients excluded could not be consented? Were they all critically ill? Language barriers?

It is possible that the 308 patients unable to be consented for this study would also not be able to be consented for HTS in the ED. If so, this study reflects a pragmatic number of patients that could be consented for HTS if the systems were in place in the ED.

Was there any data/knowledge regarding how clinicians or others determine to offer HTS to patients? Is it only clinical staff (Medical officers) that offer and order HTS, or are there separate health care workers that offer this at KNH?

Pregnant patients and prisoners are a high risk cohort and were likely excluded due to difficulty in consenting this population. Future studies may be strengthened by including these vulnerable populations.

I found just two typo/grammatical errors: In line 303, please remove the word “towards”. In line 317, please change the vernacular term, “missingness” to “missing data” or something else.

Overall, this is an extremely important research study. It will likely move the needle on ED HTS both in Kenya and around the world. I look forward to subsequent studies from your author group as you further elucidate this topic.

6. PLOS authors have the option to publish the peer review history of their article (what does this mean?). If published, this will include your full peer review and any attached files.

**Do you want your identity to be public for this peer review?** For information about this choice, including consent withdrawal, please see our Privacy Policy.

Reviewer #1: **Yes: **Eric van Praag, MD, MPH

Reviewer #2: No

---

## [Editor Report · Decision Letter 1]

26 Sep 2022

Assessment of Standard HIV Testing Services Delivery to Injured Persons Seeking Emergency Care in Nairobi, Kenya: A Prospective Observational Study

PGPH-D-22-00728R1

Dear Dr. Aluisio,

We are pleased to inform you that your manuscript 'Assessment of Standard HIV Testing Services Delivery to Injured Persons Seeking Emergency Care in Nairobi, Kenya: A Prospective Observational Study' has been provisionally accepted for publication in PLOS Global Public Health.

Best regards,

Sabine Hermans

Academic Editor